# The negative association between weight-adjusted-waist index and lung functions: NHANES 2007–2012

**Di Fan[1], Liling Zhang[2], Tingfan Wang[3]***

**1** Department of Anesthesiology, The Affiliated Hospital of Southwest Medical University, Luzhou, Sichuan Province, P. R. China, **2** Department of Nephrology, The Affiliated Hospital of Southwest Medical University, Luzhou, Sichuan Province, P. R. China, **3** Department of Pediatric Surgery, The Affiliated Hospital of Southwest Medical University, Luzhou, Sichuan Province, P. R. China

* Wangtingfan14@hotmail.com

**Data Availability Statement:** All files are available from the NHANES website(https://www.cdc.gov/nchs/nhanes/index.htm).

## Abstract

Obesity is a common public health issue worldwide, and its negative impact on lung function has garnered widespread attention. This study sought to investigate the possible association between a new obesity metric, the weight-adjusted waist index (WWI), and lung functions, providing a basis for the monitoring and protection of lung functions. We conducted a cross-sectional evaluation, analyzing data from adults in the U.S. gathered through the National Health and Nutrition Examination Survey (NHANES) from 2007 to 2012. To explore the correlation between WWIs and lung functions, we utilized a multivariate logistic regression model with appropriate weighting to ensure accuracy. Smooth curve fitting also helped to confirm the linear nature of this relationship. Subgroup analyses were conducted to confirm the uniformity and dependability of the results. Our study included data from 13,805 adults in the United States. Multivariate linear regression analysis revealed that, in the fully adjusted model, higher WWIs were negatively correlated with forced vital capacity (FVC), forced expiratory volume in the first second (FEV1), FEV1/FVC, peak expiratory flow rate (PEF), and forced expiratory flow rate (FEF) 25%-75% (β = -0.63; 95% confidence interval [CI] [-0.71, -0.55]; β = -0.55; 95% CI [-0.62, -0.48]; β = -0.02; 95% CI [-0.03, -0.01]; β = -1.44; 95% CI [-1.65, -1.23]; β = -0.52; 95% CI [-0.65, -0.39], respectively). Additionally, when analyzing the WWI as a categorical variable, a significant downward trend in the FVC, FEV1, PEF, and FEF 25%-75% was observed from Q2 to Q4 as the WWI increased (trend *P* < 0.05). Subgroup analysis showed stronger associations between WWI and lung functions, particularly among younger, non-Hispanic white, male participants, and current smokers. Our results indicate that elevated WWI is strongly associated with declining lung functions, demonstrating the importance of long-term monitoring and tracking of WWIs.

## Introduction

Obesity is an urgent public health concern and is one of the most prevalent health risks worldwide. The global incidence of obesity is rising annually because of insufficient physical exercise

**Funding:** The author(s) received no specific funding for this work.

**Competing interests:** The authors have declared that no competing interests exist.

and excessive eating [1]. More than just a significant health burden, obesity is a well-established contributing factor for a myriad of chronic conditions, including diabetes mellitus, osteoarthritis, and cardiovascular diseases [2,3]. Furthermore, obesity can compromise lung function, leading to an increased likelihood of asthma and obstructive sleep apnea syndrome [4]. Currently, waist circumference (WC) and body mass index (BMI) are the prevalent clinical indicators used to assess obesity. However, due to the "obesity paradox," relying solely on BMI or WC for assessment has its limitations. The weight-adjusted waist index (WWI), introduced by Park et al [5], is a new metric for assessing obesity. It measures the WC in centimeters and divides it by the square root of the body weight in kilograms, highlighting central obesity. A study by the same team [4] indicated that the WWI is a strong predictor of cardiometabolic disease and mortality risk. Furthermore, researchers have found that in older adults, the WWI positively correlates with body fat and inversely with muscle mass [5]. These findings suggest that the WWI, as an innovative obesity assessment tool, can accurately reflect adiposity and muscle mass components across different BMI categories, enhancing the precision of obesity classification and risk prediction. This facilitates more focused treatments and monitoring strategies.

Several studies have investigated the association between obesity and lung function [4,6,7]. Research by Park et al [4], suggested that obesity, particularly central obesity, has a significant effect on lung function in middle-aged Asians. Another study revealed that while obesity does not hinder complete inflation or deflation of the lungs, it markedly reduces the resting lung capacity in those with obesity [6]. Abdominal obesity can compromise respiratory muscle efficiency and lung compliance, influencing diaphragm and chest wall movements. This, in turn, affects the ventilation-perfusion ratios and breathing patterns, leading to diminished exercise tolerance and hypoxemia. Therefore, the need for health examination follow-up and lung function screening in individuals with abdominal obesity has been suggested to prevent the progression of chronic respiratory diseases [7]. Furthermore, oxidative stress and inflammation are believed to play crucial roles in the association between obesity and deteriorating lung functions [6,8].

Recently, the WWI has been shown to be associated with hepatic steatosis [9], abdominal aortic calcification [10] and cognitive function [11]. However, there has not been a study that examined the relationship between the WWI and lung functions. We used data from the National Health and Nutrition Examination Survey (NHANES) from 2007 to 2012 to explore the relationship between the WWI and various lung function indicators to test our hypothesis that there is a risk of diminished lung function with increasing WWIs.

## Materials and methods

### Data source and study population

The NHANES is a biennial cross-sectional survey conducted by the National Center for Health Statistics (NCHS), aimed at assessing the health and nutritional status of adults and children in the United States. The study data includes household interviews and mobile center-based laboratory tests used in epidemiological studies and health science research. NHANES uses a complex, multistage, stratified, and clustered sampling design to ensure national representativeness. In this study, we applied the appropriate sampling weights to ensure that the survey results accurately reflect the health and nutritional status of the U.S. population. Additional information is available on the NHANES website (https://www.cdc.gov/nchs/nhanes/index.htm). The survey received clearance from the NCHS Ethics Review Board, and all subjects gave their written informed agreement. The Ethics Committee of Affiliated

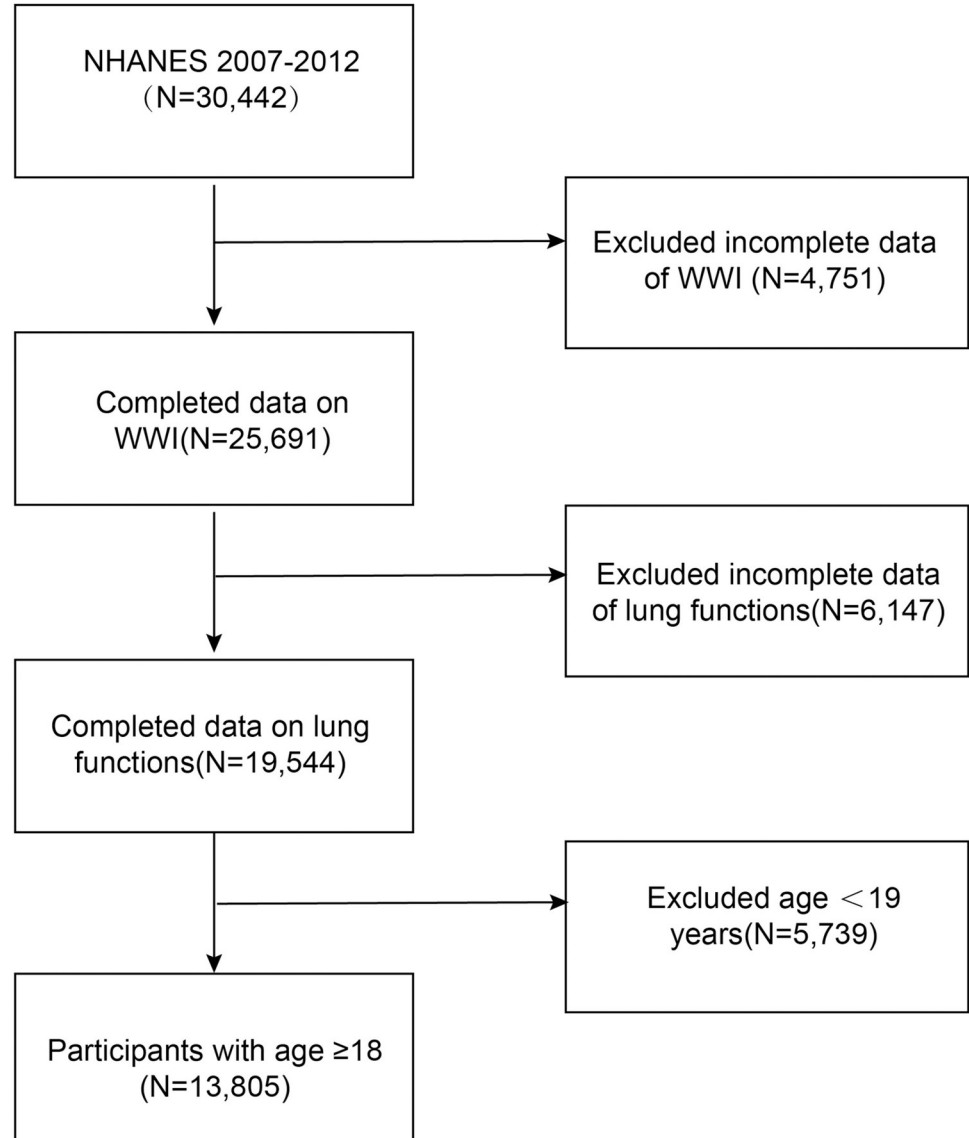

**Fig 1. Study flowchart.** WWI: Weight-adjusted waist index; NHANES: National Survey of the National Center for Health Statistics.

Hospital of Southwest Medical University has granted an exemption from review for this particular study, ethics number was KY2024148.

A total of 30,442 participants in NHANES from 2007 to 2012 were enrolled in this study. We excluded participants with missing data on the WWI (N = 4751), pulmonary function (N = 6147), and <19 years of age (N = 5739). In total, 13805 adults were selected for this study (Fig 1).

## WWI assessment

The WWI is a measurement index calculated as the patient's WC (cm) divided by the square root of their weight (kg) [5]. The "Body Measurements" information for WC (cm) and weight (kg) were obtained by skilled health technicians at the mobile examination center and were

available on the NHANES website. In this study, we selected the WWI as the exposure variable and analyzed it using both continuous and categorical approaches. Based on the WWI quartiles, we divided the survey participants into four groups (Q1-Q4).

## Lung function assessment

We selected individuals aged 19–79 years from the 2007–2012 NHANES data for spirometry tests performed according to the American Thoracic Society recommendations [12] and excluded participants who were experiencing chest pain or had problems with forceful exhalation at the time of the data collection, those on supplemental oxygen, and individuals who had recently undergone surgery on their eyes, chest, or abdomen. Furthermore, we excluded individuals who had recently experienced a heart attack or stroke, were exposed to tuberculosis, or had bloody sputum. We also omitted adults with a previous diagnosis of a dislodged retina or deflated lung, as well as children with distressing ear inflammations. Our study involved five key lung function measurements: the forced vital capacity (FVC), the forced expiratory volume in the first second (FEV1), the FEV1: FVC ratio calculated by dividing the FEV1 by the FVC, the peak expiratory flow rate (PEF), and the mid-exhalation forced expiratory flow rate (FEF 25–75%).

## Covariate assessment

Our selection of covariates was guided by earlier research on lung functions and factors related to WWIs [9–11,13,14]. Our analysis incorporated a range of variables, including age, sex, ethnicity, poverty-income ratio (PIR), weight, BMI, direct high-density and low-density lipoprotein cholesterol levels (HDL and LDL), triglyceride levels, total cholesterol levels, serum cotinine levels, smoking habits (never, former, current), alcohol intake (assessed as over or under 12 drinks per year), asthma (defined as an affirmative response to both 'Has a doctor diagnosed you with asthma?' and 'Have you experienced wheezing or whistling in your chest in the past 12 months?' with participants answering "no" to either question classified as the control group), along with diagnoses of congestive heart failure, heart attack, stroke, diabetes mellitus and hypertension(yes/no). Physical activity (PA) was also included, with PA calculated based on the MET values provided by NHANES for each type of activity, factoring in weekly frequency and duration. The formula used was PA (MET-min/wk) = MET × weekly frequency × duration of each activity. Data were collected via structured interviews, clinical evaluations, and laboratory tests facilitated by skilled healthcare personnel. Additional information is accessible on the NHANES website (https://www.cdc.gov/nchs/nhanes/index.htm).

## Statistical analysis

Analyses were carried out using R version 3.4.4 (https://www.R-project.org/), with a two-tailed *P*-value of less than 0.05 considered significant. Given that the objective of the NHANES is to reflect the U.S. non-institutionalized civilian population, our statistical methods employed NHANES-specific sample weights. To identify important confounding variables associated with lung function, we performed univariate analyses (S1 Table) and addressed multicollinearity by excluding variables with VIF values over 10 (S2 Table). We divided the participants' demographics into quartiles according to their WWIs, applying the weighted Student's t-test for continuous variables and the weighted chi-squared test for categorical variables. To examine the relationship between the WWI and lung functions, we used weighted multivariate linear regression and explored nonlinear dynamics through smooth curve fitting. Stratified and interaction analyses were also conducted to explore how this relationship varied with age, sex, race, and smoking status. The analysis framework included three models: Model 1, without

adjustment; Model 2, adjusted for age, sex, and race; and Model 3, which accounted for all covariates listed in Table 1, excluding those directly linked to the WWI and lung functions, with Model 3 serving as the basis for subgroup evaluations and curve fitting.

## Results

### Baseline characteristics of participants

Table 1 summarizes the demographic characteristics and five pulmonary function indicators of the study participants, which included 50.32% males and 49.68% females. Based on the WWI values, participants were divided into four quartiles: the ranges for the first, second, third, and fourth quartiles of WWIs were 8.11–10.32 cm/$\sqrt{}$kg, 10.32–10.91 cm/$\sqrt{}$kg, 10.91–11.48 cm/$\sqrt{}$kg, and 11.48–14.20 cm/$\sqrt{}$kg, respectively. The pulmonary function parameters of the participants decreased with an increase in WWIs ($P < 0.001$). Additionally, participants in the highest WWI quartile were more likely to be female, non-Hispanic White, smokers, have lower alcohol consumption, and have been diagnosed with asthma, coronary heart disease, congestive heart failure, stroke, hypertension and diabetes mellitus. Furthermore, with an increase in WWIs, there was a corresponding increase in age, BMI, weight, total cholesterol, triglycerides, LDL levels and physical activity. Conversely, HDL levels, PIR, and serum cotinine levels decreased with increasing WWIs (Table 1).

### Association between WWI and lung function indices

Table 2 illustrates the relationship between the WWI and lung functions using the three analytical models. In Model 1, WWIs (both continuous and categorized) showed a significant negative correlation with all lung function metrics ($P < 0.001$). With the complete adjustment in Model 3, the negative association remained robust between continuous WWIs and lung function parameters including FVC, FEV1, FEV1/FVC, PEF, and FEF 25%–75% (β = -0.63; 95% confidence interval CI [-0.71, -0.55]; β = -0.55; 95% CI [-0.62, -0.48], β = -0.02; 95% CI [-0.03, -0.01], β = -1.44; 95% CI [-1.65, -1.23], β = -0.52; 95% CI [-0.65, -0.39], respectively). Moreover, when categorizing WWIs and using the first quartile as the reference, there was a significant reduction in FVC, FEV1, PEF, and FEF 25%–75% across the second to fourth quartiles as WWIs increased ($P$ for trend < 0.05); however, the link between WWIs and FEV1/FVC was not significant ($P$ for trend = 0.336). Additionally, from a nonlinear perspective, the application of a generalized model and smooth curve fitting further corroborated this negative correlation (Fig 2).

### Subgroup analysis

The subgroup analysis (Fig 3), which explores the relationship between WWIs and lung functions, with stratification based on age, sex, race, and smoking status. The decreases in FEV1 and FVC were more strongly associated with male, non-Hispanic white, participants≤60 years of age and current smokers. Similarly, a stronger association with FEV1/FVC and FEF25%-75% were identified among current smokers and participants ≤60 years of age. Moreover, PEF showed a stronger correlation in the subgroup female, non-Hispanic white, participants>60 years of age and former smokers. These stratified analysis results suggest that the association between the WWI and lung functions may be more significant in certain populations.

## Discussion

This observational study included data from 13,805 American adults and, to our understanding, is the first study to investigate the connection between the WWI and lung functions.

**Table 1. Basic characteristics of participants by weight-adjusted waist index quartiles.**

| | Weight-adjusted waist index (cm/$\sqrt{kg}$) | | | | P-value |
|---|---|---|---|---|---|
| | Q1 (8.11–10.32) N = 3451 | Q2 (10.32–10.91) N = 3451 | Q3 (10.91–11.48) N = 3451 | Q4 (11.48–14.20) N = 3452 | |
| **Age (years)** | 34.11 ± 13.02 | 42.96 ± 13.90 | 48.34 ± 14.50 | 53.09 ± 15.65 | <0.001 |
| **Sex (%)** | | | | | <0.001 |
| **Male** | 58.83 | 54.11 | 48.49 | 34.23 | |
| **Female** | 41.17 | 45.89 | 51.51 | 65.77 | |
| **Race/ethnicity (%)** | | | | | <0.001 |
| **Mexican American** | 4.87 | 8.31 | 10.59 | 11.55 | |
| **Other Hispanic** | 4.54 | 5.64 | 6.47 | 6.04 | |
| **Non-Hispanic White** | 68.20 | 68.54 | 67.15 | 66.83 | |
| **Non-Hispanic Black** | 14.70 | 10.11 | 9.61 | 10.14 | |
| **Other Race** | 7.70 | 7.39 | 6.17 | 5.44 | |
| **PIR** | 3.02 ± 1.66 | 3.12 ± 1.63 | 2.96 ± 1.61 | 2.69 ± 1.58 | <0.001 |
| **Weight (kg)** | 73.26 ± 15.47 | 80.32 ± 19.05 | 85.85 ± 20.66 | 91.73 ± 24.07 | <0.001 |
| **BMI (kg/m$^2$)** | 24.28 ± 4.08 | 27.40 ± 5.14 | 30.13 ± 5.94 | 33.76 ± 7.36 | <0.001 |
| **HDL (mmol/L))** | 1.45 ± 0.41 | 1.37 ± 0.41 | 1.30 ± 0.38 | 1.29 ± 0.37 | <0.001 |
| **LDL (mmol/L))** | 2.84 ± 0.58 | 2.96 ± 0.57 | 2.99 ± 0.63 | 2.99 ± 0.65 | <0.001 |
| **Triglyceride (mmol/L)** | 1.16 ± 0.60 | 1.30 ± 0.80 | 1.38 ± 0.88 | 1.45 ± 0.92 | <0.001 |
| **Total Cholesterol (mmol/L)** | 4.76 ± 0.95 | 5.12 ± 0.97 | 5.23 ± 1.07 | 5.20 ± 1.10 | <0.001 |
| **Serum cotinine level (ng/mL)** | 58.28 ± 123.55 | 60.48 ± 129.65 | 54.19 ± 127.19 | 50.88 ± 118.09 | 0.009 |
| **Physical activity (MET-minutes per week)** | 3578.27 ± 4483.72 | 4432.95 ± 4728.87 | 5187.79 ± 5526.89 | 6249.58 ± 6736.21 | <0.001 |
| **Hypertension (%)** | | | | | <0.001 |
| **Yes** | 9.62 | 22.96 | 31.70 | 47.73 | |
| **No** | 90.35 | 76.98 | 68.24 | 52.10 | |
| **Don't know** | 0.03 | 0.06 | 0.06 | 0.17 | |
| **Diabetes mellitus (%)** | | | | | <0.001 |
| **Yes** | 1.59 | 3.40 | 7.90 | 18.50 | |
| **No** | 97.78 | 95.24 | 89.95 | 78.45 | |
| **Borderline** | 0.63 | 1.32 | 2.03 | 3.00 | |
| **Don't know** | | 0.04 | 0.12 | 0.05 | |
| **Asthma (%)** | | | | | <0.001 |
| **Yes** | 14.43 | 11.86 | 13.87 | 17.07 | |
| **No** | 85.57 | 88.14 | 86.13 | 82.93 | |
| **Coronary heart disease (%)** | | | | | <0.001 |
| **Yes** | 2.22 | 5.50 | 7.93 | 11.48 | |
| **No** | 97.78 | 94.50 | 92.07 | 88.52 | |
| **Congestive heart failure (%)** | | | | | <0.001 |
| **Yes** | 3.08 | 0.98 | 0.49 | 0.60 | |
| **No** | 96.92 | 99.02 | 99.51 | 99.40 | |
| **Stroke (%)** | | | | | <0.001 |
| **Yes** | 4.52 | 1.76 | 0.68 | 0.71 | |
| **Yes** | 95.48 | 98.24 | 99.32 | 99.29 | |
| **Had at least 12 alcoholic drinks/1 year? (%)** | | | | | <0.001 |
| **Yes** | 84.51 | 82.76 | 77.12 | 72.61 | |
| **No** | 15.49 | 17.24 | 22.88 | 27.39 | |
| **Smoking status (%)** | | | | | <0.001 |
| **Never smoked** | 59.66 | 56.33 | 52.48 | 50.06 | |

(*Continued*)

**Table 1.** (Continued)

| | Weight-adjusted waist index (cm/√kg) | | | | P-value |
|---|---|---|---|---|---|
| | Q1 (8.11–10.32) N = 3451 | Q2 (10.32–10.91) N = 3451 | Q3 (10.91–11.48) N = 3451 | Q4 (11.48–14.20) N = 3452 | |
| **Former smoker** | 16.93 | 21.06 | 28.05 | 28.94 | |
| **Current smoker** | 23.42 | 22.61 | 19.48 | 21.00 | |
| **WWI (cm/√kg)** | 9.85 ± 0.36 | 10.63 ± 0.17 | 11.18 ± 0.17 | 11.96 ± 0.39 | <0.001 |
| **FEV1 (L)** | 3.77 ± 0.87 | 3.38 ± 0.82 | 3.06 ± 0.77 | 2.59 ± 0.75 | <0.001 |
| **FVC (L)** | 4.70 ± 1.06 | 4.33 ± 1.01 | 3.95 ± 0.94 | 3.37 ± 0.90 | <0.001 |
| **FEV1/FVC** | 0.80 ± 0.08 | 0.78 ± 0.08 | 0.78 ± 0.08 | 0.77 ± 0.09 | <0.001 |
| **PEF (L/s)** | 9.13 ± 2.10 | 8.66 ± 2.12 | 8.12 ± 2.07 | 7.00 ± 1.96 | <0.001 |
| **FEF 25%-75% (L/s)** | 3.66 ± 1.30 | 3.16 ± 1.22 | 2.85 ± 1.21 | 2.40 ± 1.18 | <0.001 |

Mean ± SD for continuous variables; P-values were calculated using the weighted linear regression model; (%) for categorical variables and P-values were calculated using the weighted chi-squared test for categorical variables. Q: Quartile; PIR: The ratio of income to poverty; BMI: Body mass index; HDL: High-density lipoprotein; LDL: Low density lipoprotein; WWI: Weight-adjusted waist index; FEV1: Forced expiratory volume in 1 s; FVC: Forced vital capacity; FEV1/FVC: Ratio of FEV1 to FVC; PEF: Peak expiratory flow rate; FEF 25–75%: Forced expiratory flow between 25 and 75% of FVC.

Initially, we found a negative correlation between the WWI and all lung function indicators. After fully adjusting for covariates, we noted that an increase in the WWI was closely linked to decreases in the FVC, FEV1, PEF%, and FEF25%–75%. This negative correlation persisted when WWIs were divided into quartiles (Q1–Q4), and nonlinear fitting supported our hypothesis. Further analysis revealed that the association between the WWI and lung functions was more pronounced in younger, current smokers, non-Hispanic white, and male participants. This may be due to our method of using actual lung volume measurements instead of predicted percentages and introducing confounding factors such as age, sex, and race [15]. These methodological enhancements are essential for future research. Our results provide evidence of a negative correlation between the WWI and lung functions, highlighting the potential impact of the WWI on pulmonary health and emphasizing the importance of monitoring and mitigating high WWIs for lung well-being.

Obesity is widely recognized as a critical health challenge in Western nations and serves as a catalyst for numerous health issues and adverse outcomes. An increase in abdominal fat is associated with the onset of hypertension, diabetes mellitus, and other facets of metabolic syndrome [16]. Additionally, weight gain amplifies mechanical stress, potentially leading to osteoarthritis [17] and discomfort in the back and lower extremities [18]. The persistent, low-grade inflammation observed in adipose tissue contributes to metabolic disorders and complications in organ systems within the obese demographic [19]. Numerous studies have supported the link between obesity and impaired lung functions. For instance, research involving South African adolescents of African ancestry demonstrated a negative correlation between their BMIs and FEV1/FVC ratios, indicating a potential association between obesity and airway obstruction in this group [20]. Moreover, a systematic review [21] elaborated on how childhood obesity influences lung capacity, mechanics, airway function, and exercise capacity, revealing that excess weight can negatively impact both static and dynamic respiratory functions to varying degrees. Further extending the research population, a cohort study [22] in the Chinese population indicated a U-shaped relationship between lung functions and the visceral adiposity index, with both very low and very high VAI levels being closely associated with reduced lung functions. Another study [23] within the Chinese demographic showed that both underweight and severe obesity are linked to diminished lung functions, while mild obesity appears to serve

**Table 2. The association between WWI and lung function.**

| | Crude model β (95% CI) P-value | Minimally adjusted model β (95% CI) P-value | Fully adjusted model β (95% CI) P-value |
|---|---|---|---|
| **FVC** | | | |
| **WWI (continuous)** | -0.61 (-0.63, -0.59) * | -0.30 (-0.31, -0.28) * | -0.63 (-0.71, -0.55) * |
| **WWI groups** | | | |
| **Quartile 1** | Reference | Reference | Reference |
| **Quartile 2** | -0.37 (-0.41, -0.33) * | -0.14 (-0.17, -0.11) * | -0.10 (-0.13, -0.07) * |
| **Quartile 3** | -0.75 (-0.80, -0.70) * | -0.34 (-0.37, -0.30) * | -0.21 (-0.25, -0.16) * |
| **Quartile 4** | -1.33 (-1.37, -1.28) * | -0.63 (-0.66, -0.59) * | -0.33 (-0.40, -0.26) * |
| **P for trend** | <0.001 | <0.001 | <0.001 |
| **FEV1** | | | |
| **WWI (continuous)** | -0.55 (-0.57, -0.53) * | -0.23 (-0.24, -0.22) * | -0.55 (-0.62, -0.48) * |
| **WWI group** | | | |
| **Quartile 1** | Reference | Reference | Reference |
| **Quartile 2** | -0.39 (-0.43, -0.35)* | -0.13 (-0.16, -0.11) * | -0.09 (-0.12, -0.06) * |
| **Quartile 3** | -0.71 (-0.75, -0.67) * | -0.26 (-0.29, -0.24) * | -0.16 (-0.20, -0.12)* |
| **Quartile 4** | -1.18 (-1.22, -1.14) * | -0.48 (-0.51, -0.45) * | -0.25 (-0.31, -0.19)* |
| *P* for trend | <0.001 | <0.001 | <0.001 |
| **FEV1/FVC** | | | |
| **WWI (continuous)** | -0.02 (-0.02, -0.02) * | 0.00 (-0.00, 0.00) | -0.02 (-0.03, -0.01) * |
| **WWI group** | | | |
| **Quartile 1** | Reference | Reference | Reference |
| **Quartile 2** | -0.02 (-0.03, -0.02)* | -0.00 (-0.00, 0.00) | -0.00 (-0.01, 0.00) |
| **Quartile 3** | -0.03 (-0.03, -0.02) * | 0.00 (-0.00, 0.01) | -0.00 (-0.01, 0.00) |
| **Quartile 4** | -0.04 (-0.04, -0.03) * | 0.00 (-0.00, 0.01) | -0.00 (-0.01, 0.00) |
| *P* for trend | <0.001 | 0.097 | 0.336 |
| **PEF** | | | |
| **WWI (continuous)** | -0.97 (-1.02, -0.93) * | -0.41 (-0.45, -0.37) * | -1.44 (-1.65, -1.23) * |
| **WWI group** | | | |
| **Quartile 1** | Reference | Reference | Reference |
| **Quartile 2** | -0.47 (-0.56, -0.38) * | -0.08 (-0.15, -0.00) * | -0.13 (-0.22, -0.04) * |
| **Quartile 3** | -1.01 (-1.11, -0.91) * | -0.30 (-0.38, -0.22) * | -0.31 (-0.43, -0.18) * |
| **Quartile 4** | -2.13 (-2.23, -2.03) * | -0.90 (-0.99, -0.82) * | -0.75 (-0.94, -0.57) * |
| *P* for trend | <0.001 | <0.001 | <0.001 |
| **FEF25%-75%** | | | |
| **WWI (continuous)** | -0.60 (-0.62, -0.57) * | -0.13 (-0.15, -0.11) * | -0.52 (-0.65, -0.39) * |
| **WWI group** | | | |
| **Quartile 1** | Reference | Reference | Reference |
| **Quartile 2** | -0.50 (-0.56, -0.45) * | -0.08 (-0.13, -0.04) * | -0.07 (-0.13, -0.02) * |
| **Quartile 3** | -0.81 (-0.87, -0.76) * | -0.12 (-0.17, -0.07)* | -0.08 (-0.16, -0.01) * |
| **Quartile 4** | -1.27 (-1.33, -1.21) * | -0.27 (-0.32, -0.21) * | -0.18 (-0.29, -0.06) * |
| *P* for trend | <0.001 | <0.001 | 0.005 |

*$p<0.05$. Values are presented as β (95% confidence interval) or *P*-values. Crude model: No covariates were adjusted. The minimally adjusted model was adjusted for age, sex, race, and ethnicity. Fully adjusted model: Except for the stratification component, all covariates presented in Table 1 were adjusted (age, sex, race/ethnicity, BMI, PIR, weight, HDL, LDL, triglyceride, total cholesterol, serum cotinine levels, physical activity, smoking status, alcohol consumption, asthma, coronary heart disease, congestive heart failure, stroke, diabetes mellitus, and hypertension). CI: Confidence interval; FEV1: Forced expiratory volume in 1 s; FVC: Forced vital capacity; FEV1/FVC: Ratio in FEV1 to FVC; PEF: Peak expiratory flow rate; FEF 25–75%: Forced expiratory flow in 25 and 75% of FVC.

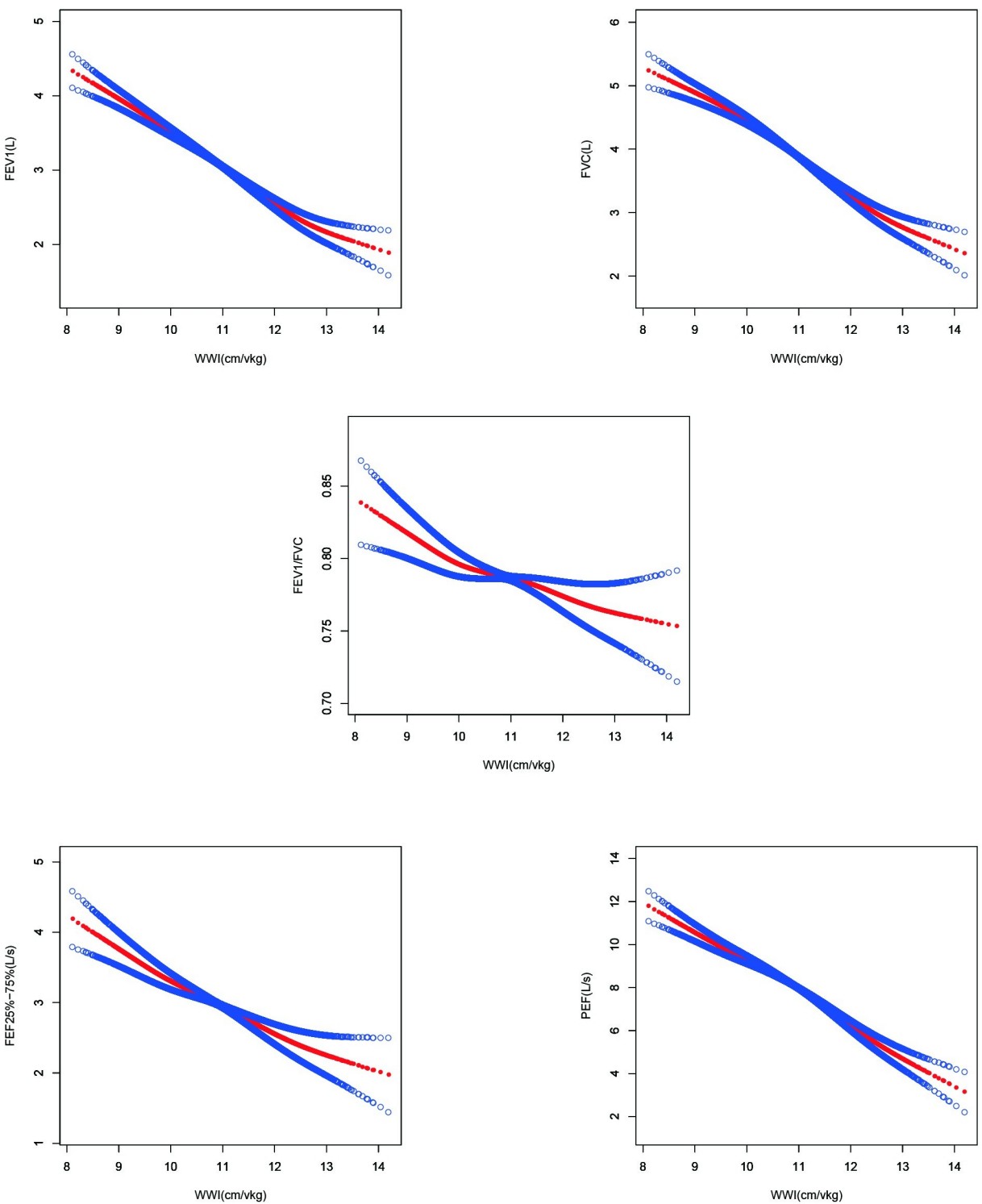

**Fig 2. Association between WWI and lung functions for Model III.** Adjusted for age, sex, race/ethnicity, BMI, PIR, weight, HDL, LDL, triglyceride, total cholesterol, serum cotinine levels, physical activity, smoking status, alcohol consumption, asthma, coronary heart disease, congestive heart failure, stroke, diabetes mellitus, and hypertension. The solid crimson line signifies the smooth curve fitting among the variables. Azure bands indicate the 95% confidence interval stemming from the fit. WWI: Weight-adjusted waist index; FEV1: Forced expiratory volume in 1 s; FVC: Forced vital capacity; FEV1/FVC: Ratio of FEV1 to FVC; PEF: Peak expiratory flow rate; FEF 25–75%: Forced expiratory flow between 25 and 75% of FVC.

## FEV1(L) and WWI(cm/√kg)

| Subgroups | β(95% CI) | | P for interaction |
|---|---|---|---|
| Gender | | | <0.001 |
| Male | -0.84 (-0.91 , -0.77) | | |
| Female | -0.58 (-0.65 , -0.51) | | |
| Age(years) | | | <0.001 |
| ≤60 | -0.76 (-0.84 , -0.69) | | |
| >60 | -0.62 (-0.69 , -0.54) | | |
| Race | | | <0.001 |
| Mexican American | -0.60 (-0.69 , -0.52) | | |
| Other Hispanic | -0.62 (-0.72 , -0.53) | | |
| Non-Hispanic White | -0.74 (-0.81 , -0.66) | | |
| Non-Hispanic Black | -0.55 (-0.64 , -0.46) | | |
| Other Races | -0.67 (-0.75 , -0.58) | | |
| Smoke status | | | <0.001 |
| Never smoke | -0.66 (-0.73 , -0.58) | | |
| Former smoke | -0.71 (-0.79 , -0.63) | | |
| Current smoke | -0.80 (-0.88 , -0.73) | | |

negative correlation   positive correlation

## FVC(L) and WWI(cm/√kg)

| Subgroups | β(95% CI) | | P for interaction |
|---|---|---|---|
| Gender | | | <0.001 |
| Male | -0.77 (-0.86 , -0.67) | | |
| Female | -0.45 (-0.53 , -0.37) | | |
| Age(years) | | | <0.001 |
| ≤60 | -0.81 (-0.90 , -0.73) | | |
| >60 | -0.71 (-0.80 , -0.63) | | |
| Race | | | 0.018 |
| Mexican American | -0.53 (-0.80 , -0.27) | | |
| Other Hispanic | -0.14 (-0.48 , 0.20) | | |
| Non-Hispanic White | -0.80 (-0.90 , -0.70) | | |
| Non-Hispanic Black | -0.34 (-0.56 , -0.13) | | |
| Other Races | 0.05 (-0.26 , 0.35) | | |
| Smoke status | | | <0.001 |
| Never smoke | -0.61 (-0.69 , -0.53) | | |
| Former smoke | -0.66 (-0.75 , -0.58) | | |
| Current smoke | -0.73 (-0.81 , -0.65) | | |

negative correlation   positive correlation

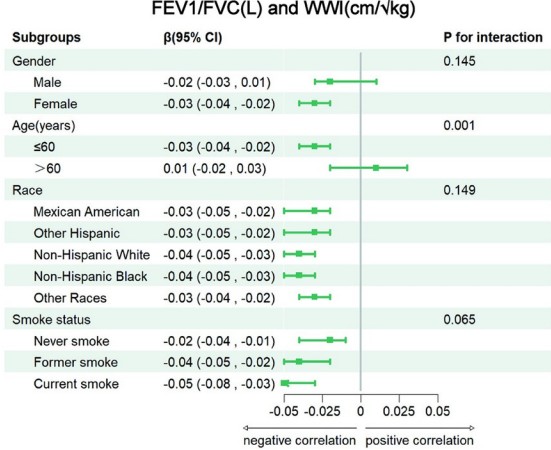

### FEV1/FVC(L) and WWI(cm/√kg)

| Subgroups | β(95% CI) | | P for interaction |
|---|---|---|---|
| Gender | | | 0.145 |
| Male | -0.02 (-0.03 , 0.01) | | |
| Female | -0.03 (-0.04 , -0.02) | | |
| Age(years) | | | 0.001 |
| ≤60 | -0.03 (-0.04 , -0.02) | | |
| >60 | 0.01 (-0.02 , 0.03) | | |
| Race | | | 0.149 |
| Mexican American | -0.03 (-0.05 , -0.02) | | |
| Other Hispanic | -0.03 (-0.05 , -0.02) | | |
| Non-Hispanic White | -0.04 (-0.05 , -0.03) | | |
| Non-Hispanic Black | -0.04 (-0.05 , -0.03) | | |
| Other Races | -0.03 (-0.04 , -0.02) | | |
| Smoke status | | | 0.065 |
| Never smoke | -0.02 (-0.04 , -0.01) | | |
| Former smoke | -0.04 (-0.05 , -0.02) | | |
| Current smoke | -0.05 (-0.08 , -0.03) | | |

negative correlation   positive correlation

## PEF(L/s) and WWI(cm/√kg)

| Subgroups | β(95% CI) | | P for interaction |
|---|---|---|---|
| Gender | | | <0.001 |
| Male | -1.68 (-1.89 , -1.46) | | |
| Female | -1.70 (-1.93 , -1.48) | | |
| Age(years) | | | 0.002 |
| ≤60 | -1.56 (-1.77 , -1.35) | | |
| >60 | -1.70 (-1.93 , -1.48) | | |
| Race | | | 0.003 |
| Mexican American | -1.45 (-1.69 , -1.21) | | |
| Other Hispanic | -1.35 (-1.59 , -1.10) | | |
| Non-Hispanic White | -1.52 (-1.74 , -1.31) | | |
| Non-Hispanic Black | -1.41 (-1.64 , -1.19) | | |
| Other Races | -0.43 (-1.35 , 0.48) | | |
| Smoke status | | | <0.001 |
| Never smoke | -1.52 (-1.73 , -1.31) | | |
| Former smoke | -1.86 (-2.08 , -1.63) | | |
| Current smoke | -1.72 (-1.94 , -1.50) | | |

negative correlation   positive correlation

## FEF25%-75%(L/s)and WWI(cm/√kg)

| Subgroups | β(95% CI) | | P for interaction |
|---|---|---|---|
| Gender | | | 0.078 |
| Male | -1.00 (-1.25 , -0.75) | | |
| Female | -0.73 (-0.91 , -0.54) | | |
| Age(years) | | | <0.001 |
| ≤60 | -0.96 (-1.11 , -0.80) | | |
| >60 | -0.24 (-0.58 , 0.10) | | |
| Race | | | 0.460 |
| Mexican American | -0.36 (-0.81 , 0.09) | | |
| Other Hispanic | -0.33 (-0.90 , 0.25) | | |
| Non-Hispanic White | -0.64 (-0.81 , -0.47) | | |
| Non-Hispanic Black | -0.27 (-0.64 , 0.09) | | |
| Other Races | -0.13 (-0.65 , 0.38) | | |
| Smoke status | | | <0.001 |
| Never smoke | -0.70 (-0.84 , -0.56) | | |
| Former smoke | -0.82 (-0.97 , -0.66) | | |
| Current smoke | -0.96 (-1.11 , -0.81) | | |

negative correlation   positive correlation

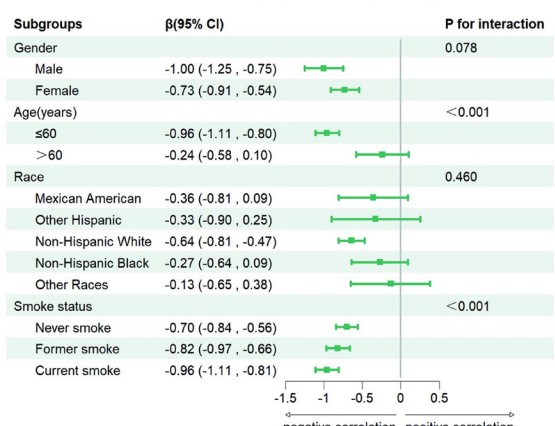

**Fig 3. Subgroup analysis for the association between WWI and lung functions for Model III.** Adjusted for age, sex, race/ethnicity, BMI, PIR, weight, HDL, LDL, triglyceride, total cholesterol, serum cotinine levels, physical activity, smoking status, alcohol consumption, asthma, coronary heart disease, congestive heart failure, stroke, diabetes mellitus, and hypertension. CI: Confidence interval. WWI: Weight-adjusted waist index; FEV1: Forced expiratory volume in 1 s; FVC: Forced vital capacity; FEV1/FVC: Ratio of FEV1 to FVC; PEF: Peak expiratory flow rate; FEF 25–75%: Forced expiratory flow between 25 and 75% of FVC.

as a protective factor for lung functions in patients at risk for COPD. These findings highlight the importance of evaluating obesity levels in predicting lung function trends.

Many studies [5,24–26] have shown that traditional metrics, such as the BMI and WC, fall short as they do not differentiate between fat and muscle mass, leading to the so-called "obesity paradox" and compromising result accuracy. One study [27] demonstrated that abdominal fat is a more reliable indicator of lung health than the overall weight or BMI. Chen et al [28] observed a significant inverse relationship between waist size and lung capacity in individuals with normal weight, overweight, and obesity, a pattern that does not hold when only the BMI is considered. Furthermore, another study [29] found a connection between a higher body fat percentage and reduced respiratory functions in adults. However, it also highlighted that basic measures of obesity, such as body mass or BMI, fail to fully capture how the composition and distribution of body fat impact lung functions. The WWI proposed by Park et al [5], represents a more accurate and comprehensive indicator of obesity. This straightforward calculation method offers robust predictive power for disease progression. Thus, the WWI holds significant promise as a potential anthropometric measure. Given the established connection between obesity and lung functions, we hypothesized a close relationship between the WWI and pulmonary health, which is corroborated by our research.

The mechanisms by which the WWI affects lung functions have not yet been fully elucidated; however, several theories have been proposed. First, fat accumulation in the mediastinum and abdominal and thoracic cavities directly affects the mechanical properties of the lungs and chest wall. This build-up restricts the movement of the diaphragm and increases pressure in the pleural cavity, which, in turn, affects lung functions [30,31]. Second, the pronounced impact of obesity on lung functions might be attributed to altered signaling molecules from fat cells, significantly influencing the onset and regulation of inflammation. This leads to chronic low-grade inflammation that either directly or indirectly affects the lungs [8,19,32,33]. Additionally, changes in the inflammatory cytokines from fat, such as tumor necrosis factor-alpha, leptin, and adiponectin, could increase airway sensitivity and elevate the risk of lung conditions such as asthma [34]. Lastly, there is a suggested link between insulin resistance and lung diseases. One study indicated that poor diabetes mellitus management correlated with reduced lung function in cystic fibrosis [35], whereas data from both mice and humans suggest that insulin resistance may play a role in the development of pulmonary arterial hypertension [36].

Our research has several strengths; notably, it is the first of its kind to explore the connection between the WWI and lung functions, to the best of our knowledge. We incorporated a substantial, nationally representative cohort of adults and made adjustments for a range of confounding factors to confirm the solidity of our results. Subgroup analyses were performed to examine the stability of the relationship between the WWI and lung functions in diverse populations. Nonetheless, unraveling the complex relationship between the WWI and lung functions remains challenging. It is crucial to acknowledge that we were unable to observe any potential causal links between increased WWIs and impaired lung functions in adults. Further experimental and longitudinal studies are needed to validate these observations. Moreover, despite incorporating several variables, we could not eliminate all potential effects owing to confounding factors. The inclusion of additional covariates in future studies and the use of advanced methodologies such as mediation analysis or propensity score matching may help further clarify the influence of confounders and strengthen causal inference in this area.

## Conclusions

This study revealed a correlation between the risk of diminishing lung functions and increased WWIs, underscoring the significance of early identification and prevention of

lung function deterioration through clinical surveillance of WWIs and proactive weight control.

## Supporting information

**S1 Checklist. STROBE statement—Checklist of items that should be included in reports of *cohort studies*.**
(DOC)

**S1 Table. Univariate analysis of variables associated with lung functions.**
(DOCX)

**S2 Table. Generalized variance inflation factor (GVIF) analysis results.**
(DOCX)

## Acknowledgments

We would like to thank Editage (www.editage.cn) for the English language editing.

## Author Contributions

**Conceptualization:** Di Fan, Tingfan Wang.

**Data curation:** Di Fan, Liling Zhang.

**Formal analysis:** Di Fan, Liling Zhang.

**Funding acquisition:** Tingfan Wang.

**Investigation:** Di Fan.

**Methodology:** Tingfan Wang.

**Resources:** Liling Zhang.

**Supervision:** Tingfan Wang.

**Validation:** Liling Zhang.

**Visualization:** Di Fan, Liling Zhang.

**Writing – original draft:** Di Fan.

**Writing – review & editing:** Tingfan Wang.

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
