## [Decision Letter · Decision Letter 0]

6 Aug 2024

PONE-D-24-22288The negative association between weight-adjusted-waist index and lung functions: NHANES 2007-2012PLOS ONE

Dear Dr. Wang,

Thank you for submitting your manuscript to PLOS ONE. After careful consideration, we feel that it has merit but does not fully meet PLOS ONE’s publication criteria as it currently stands. Therefore, we invite you to submit a revised version of the manuscript that addresses the points raised during the review process.

ACADEMIC EDITOR: The submitted paper (PONE-D-24-22288: The negative association between weight-adjusted-waist index and lung functions: NHANES 2007-2012) has been reviewed by the original reviewers, who have provided critical comments that the authors need to address in the revised version. 

Best  Please submit your revised manuscript by Sep 20 2024 11:59PM. If you will need more time than this to complete your revisions, please reply to this message or contact the journal office at plosone@plos.org. Please include the following items when submitting your revised manuscript:A rebuttal letter that responds to each point raised by the academic editor and reviewer(s). You should upload this letter as a separate file labeled 'Response to Reviewers'.A marked-up copy of your manuscript that highlights changes made to the original version. You should upload this as a separate file labeled 'Revised Manuscript with Track Changes'.An unmarked version of your revised paper without tracked changes. You should upload this as a separate file labeled 'Manuscript'.

We look forward to receiving your revised manuscript.

Kind regards,

Zahra Cheraghi, Ph.D

Academic Editor

PLOS ONE

Journal Requirements:

2. Please upload a copy of Figures 1,2,and 3, to which you refer in your text on pages 5 and 16. If the figure is no longer to be included as part of the submission please remove all reference to it within the text.

Reviewers' comments:

Reviewer's Responses to Questions

**Comments to the Author**

1. Is the manuscript technically sound, and do the data support the conclusions?

Reviewer #1: Yes

Reviewer #2: Yes

2. Has the statistical analysis been performed appropriately and rigorously? 

Reviewer #1: Yes

Reviewer #2: Yes

3. Have the authors made all data underlying the findings in their manuscript fully available?

Reviewer #1: Yes

Reviewer #2: Yes

4. Is the manuscript presented in an intelligible fashion and written in standard English?

Reviewer #1: No

Reviewer #2: Yes

5. Review Comments to the Author

Reviewer #1: Dear Authors,

I Read the manuscript interestingly and I have some comments as below:

1. The link to access the data used from NHANES should be reported in the manuscript.

2. In the methodology, standard protocols for using NHANES data should be reported.

3. On page 4, the first line says that "several studies..." but no reference is reported for it.

4. Provide a reference for how to calculate WWI.

5. To adjust the confounding variables, the most important step is to correctly identify these variables (confounders). There are different methods to identify the confounding variable in a relationship. One of these methods is using a DAG (Directed Acyclic Graph). Your criteria in this study to identify confounding variables are not clear.

6. The way of designing the table and reporting the results in the table (especially Table 2) is not very suitable.

7. In addition to the previous comment about how to identify confounding variables, there are many other variables that play a serious confounding role in this case and they have not been mentioned.

Reviewer #2: Dear author

The manuscript is technically sound, and the data support the conclusion.

The statistical analysis has been performed appropriately and rigorously.

I think the authors have made all data underlying the findings in their manuscript fully available, however if the data of history of asthma was exactly exist, the basic idea of the paper could be evaluated and assessed with more accuracy.

The manuscript is presented in an intelligible fashion and written in standard English.

6. PLOS authors have the option to publish the peer review history of their article (what does this mean?). If published, this will include your full peer review and any attached files.

Reviewer #1: **Yes: **Amir Almasi-Hashiani

Reviewer #2: No

---

## [Author Response · Author response to Decision Letter 0]

10 Sep 2024

September 9, 2024

Ref: PONE-D-24-22288.

Dear Editors and Reviewers,

Thank you very much for reviewing our manuscript. We greatly appreciate the constructive comments and suggestions you provided. Attached is the revised manuscript titled “The negative association between weight-adjusted-waist index and lung functions: NHANES 2007-2012”, We have carefully addressed each of the review comments. Please refer to the accompanying detailed response to reviewers.

Thank you once again for your time and consideration of our manuscript.

Sincerely yours,

Tingfan Wang, M.D.

Department of Pediatric Surgery

Affiliated Hospital of Southwest Medical University

Point-by-point response to the editor and reviewers' comments

Editor comments

1.When submitting your revision, we need you to address these additional requirements. Please ensure that your manuscript meets PLOS ONE's style requirements, including those for file naming. 

Response: Thank you for the additional requirements. we have made the article revisions according to the journal's requirements, including file naming conventions.

2. Please upload a copy of Figures 1,2,and 3, to which you refer in your text on pages 5 and 16. If the figure is no longer to be included as part of the submission please remove all reference to it within the text.

Response: Thank you for your feedback. Regarding this issue, in the previous version of the manuscript, I mistakenly uploaded Figures 1-3 as supplementary files. In the revised version, I have corrected this mistake and have now properly uploaded Figures 1-3 as part of the main submission.

Reviewers' comments:

Reviewer #1: 

1.The link to access the data used from NHANES should be reported in the manuscript.

Response: Thank you for your suggestions. We have added the link to access the NHANES data to ensure transparency and reproducibility (Page 5, Line 97). The data can be accessed through the following link: National Health and Nutrition Examination Survey (NHANES): https://www.cdc.gov/nchs/nhanes/index.htm.

2. In the methodology, standard protocols for using NHANES data should be reported.

Response: Thank you for your valuable comment. In this study, we followed the standard protocols for using NHANES data as outlined by the National Center for Health Statistics (NCHS). This includes the use of survey design variables, such as strata and clusters, as well as the application of sampling weights to ensure national representativeness. We have also adhered to all ethical guidelines provided by NCHS, including obtaining informed consent from all participants.

We have now included a more detailed description of these protocols in the methodology section of the revised manuscript (Page 5, Line 93-101). 

3. On page 4, the first line says that "several studies..." but no reference is reported for it.

Response: Thank you for pointing this out. We have now included appropriate references to support the statement on page 4 that "several studies...". These references have been added in the revised manuscript (Page 4, Line 67-68).

4. Provide a reference for how to calculate WWI.

Response: Thank you for your valuable suggestion. In the revised manuscript, we have added the appropriate reference to support the calculation method of the Waist-to-Weight Index (WWI) (Page 6, Line 112). Please refer to the updated manuscript for more details.

5. To adjust the confounding variables, the most important step is to correctly identify these variables (confounders). There are different methods to identify the confounding variable in a relationship. One of these methods is using a DAG (Directed Acyclic Graph). Your criteria in this study to identify confounding variables are not clear.

Response: Thank you for your valuable comments on the confounder identification methods in our study. We fully understand the importance of Directed Acyclic Graphs (DAG) in causal inference, as it is a powerful tool for identifying confounding factors. However, considering the cross-sectional design of our study and its specific context, we have opted for a method that is more suitable for the current research question, primarily based on existing literature, univariate analysis, and collinearity analysis to identify confounding variables.

First, we systematically reviewed the literature on the relationship between lung function and various demographic, behavioral, and health-related factors, identifying the following categories of factors that could potentially serve as confounders:

1.Obesity and Fat Metabolism: The literature suggests that obesity indicators (such as weight, BMI, and waist circumference) are closely related to lung function[1, 2].

2. Smoking and Alcohol Consumption: Smoking and drinking habits are widely recognized to affect lung function[3, 4].

3. Cardiovascular Diseases: The presence of heart disease, coronary artery disease, and other related conditions may impact lung function[5, 6].

4. Asthma and Physical Activity: These factors have been shown to be potentially associated with lung function[7, 8].

Based on this evidence, we identified the following variables as potential confounders: age, gender, race, poverty-income ratio (PIR), weight, waist circumference (WC), BMI, high-density lipoprotein (HDL) and low-density lipoprotein (LDL), triglyceride (TG) levels, total cholesterol levels, serum cotinine levels, smoking status (never, former, current), alcohol intake (more than 12 drinks per year or less), as well as diagnoses of hypertension and diabetes. Following a review of the comments and further examination of relevant literature, we decided to include several additional covariates: asthma, physical activity, Congestive heart failure, coronary artery disease, and stroke.

To ensure the validity of the confounding factors, we also conducted a series of univariate Analysis. For each dependent variable, we performed univariate regression analysis to identify confounding variables that were significant for at least one dependent variable, ensuring that key influencing factors were included. Then, we assessed potential collinearity issues by calculating the Variance Inflation Factor (VIF) and excluded variables with VIF values greater than 10 (such as waist circumference) to avoid multicollinearity problems in the model and enhance the reliability of the model estimates.

By selecting confounding factors based on existing literature and combining this with appropriate statistical analyses, we believe this approach effectively addresses potential confounders in the study and reduces possible bias. Although DAG indeed has its advantages in identifying confounding factors, we believe the current method is more suitable for the specific circumstances of this study and, with broad support from the literature, provides a solid foundation for our analysis.

References

1. Park Y, Kim J, Kim YS, Leem AY, Jo J, Chung K, et al. Longitudinal association between adiposity changes and lung function deterioration. Respiratory research. 2023;24(1):44.

2. Salome CM, King GG, Berend N. Physiology of obesity and effects on lung function. Journal of applied physiology (Bethesda, Md : 1985). 2010;108(1):206-211.

3. Thacher JD, Schultz ES, Hallberg J, Hellberg U, Kull I, Thunqvist P, et al. Tobacco smoke exposure in early life and adolescence in relation to lung function. The European respiratory journal. 2018;51(6)

4. Sisson JH. Alcohol and airways function in health and disease. Alcohol (Fayetteville, NY). 2007;41(5):293-307.

5. Higbee DH, Granell R, Sanderson E, Davey Smith G, Dodd JW. Lung function and cardiovascular disease: a two-sample Mendelian randomisation study. The European respiratory journal. 2021;58(3)

6. Zhang J, Gong Z, Li R, Gao Y, Li Y, Li J, et al. Influence of lung function and sleep-disordered breathing on stroke: a community-based study. European journal of neurology. 2018;25(11):1307-e1112.

7. Betancor D, Olaguibel JM, Rodrigo-Muñoz JM, Alvarez Puebla MJ, Arismendi E, Barranco P, et al. Lung Function Abnormalities and Their Correlation With Clinical Characteristics and Inflammatory Markers in Adult Asthma. Journal of investigational allergology & clinical immunology. 2023;33(4):294-296.

8. Bédard A, Carsin AE, Fuertes E, Accordini S, Dharmage SC, Garcia-Larsen V, et al. Physical activity and lung function-Cause or consequence? PloS one. 2020;15(8):e0237769.

6. The way of designing the table and reporting the results in the table (especially Table 2) is not very suitable.

Response: Thank you for your valuable suggestion regarding the design and reporting format of the tables. In response, we have not only revised Table 2 but also made comprehensive adjustments to all the tables in the manuscript to improve their clarity and suitability.

7. In addition to the previous comment about how to identify confounding variables, there are many other variables that play a serious confounding role in this case and they have not been mentioned.

Response: Thank you for your valuable comments. We acknowledge that there may be other variables in our analysis that could play a confounding role. In our study, we focused on identifying and adjusting for confounding factors based on a literature review, which included variables such as age, sex, race, poverty income ratio (PIR), body weight, BMI, cholesterol levels, smoking status, alcohol intake, as well as hypertension and diabetes. Additionally, after a further extensive review of the relevant literature, we also included asthma, physical activity, heart disease, coronary artery disease, stroke, and heart failure to further adjust for confounding factors. However, we recognize that, despite our best efforts, there may still be other confounding variables that were not included in our analysis. Identifying and controlling for all potential confounding factors is a challenging task, particularly in observational studies.

To address this limitation, we discussed in the discussion section that future research may benefit from incorporating additional covariates such as environmental exposures, socioeconomic status, and genetic predispositions, which may also affect the relationship between WWI and lung function. Additionally, the use of advanced methods such as mediation analysis and propensity score matching could further clarify the influence of confounding factors and strengthen causal inference in this area (Page 21, Line 327-335).

Reviewer #2:

1.The manuscript is technically sound, and the data support the conclusion. The statistical analysis has been performed appropriately and rigorously. I think the authors have made all data underlying the findings in their manuscript fully available, however if the data of history of asthma was exactly exist, the basic idea of the paper could be evaluated and assessed with more accuracy. The manuscript is presented in an intelligible fashion and written in standard English.

Response: Thank you for your positive feedback and for acknowledging the technical soundness and clarity of our manuscript. We also appreciate your suggestion regarding the inclusion of asthma history data. In the revised manuscript, we have included asthma as a covariate. The asthma information was collected using a self-administered questionnaire, completed during clinic visits. Individuals with current asthma were defined as those who provided affirmative responses to both of the following questions: ‘Has a doctor or other health professional ever told you that you have asthma?’ and ‘In the past 12 months, have you (or has the participant) had wheezing or whistling in the chest?’. Control subjects were defined as participants without current asthma who answered ‘NO’ to either question. Please review the revised manuscript at your convenience (Page 7, Line 138-141).

.

---

## [Editor Report · Decision Letter 1]

23 Sep 2024

The negative association between weight-adjusted-waist index and lung functions: NHANES 2007-2012

PONE-D-24-22288R1

Dear Dr. Wang,

We’re pleased to inform you that your manuscript has been judged scientifically suitable for publication and will be formally accepted for publication once it meets all outstanding technical requirements.

Kind regards,

Zahra Cheraghi, Ph.D

Academic Editor

PLOS ONE

Additional Editor Comments (optional):

Dear Editor

I hope this message finds you well. I am writing to formally accept the invitation to review the manuscript titled “PONE-D-24-22288R1

: The negative association between weight-adjusted-waist index and lung functions: NHANES 2007-2012],” submitted to [Journal Name]. It is an honor to contribute to the review process of this journal, and I appreciate the opportunity to engage with the important work being conducted in our field.

My final decision, based on the original reviewers' opinions, is acceptance.

Thank you once again for this opportunity. I look forward to contributing to the advancement of our discipline through this review process.

Best regards,
---

## [Editor Report · Acceptance letter]

12 Oct 2024

PONE-D-24-22288R1 

PLOS ONE

Dear Dr. Wang, 

I'm pleased to inform you that your manuscript has been deemed suitable for publication in PLOS ONE. Congratulations! Your manuscript is now being handed over to our production team.

Kind regards, 

on behalf of

Dr. Zahra Cheraghi 

Academic Editor

PLOS ONE